# The Yin and Yang of Breast Cancer: Ion Channels as Determinants of Left–Right Functional Differences

**DOI:** 10.3390/ijms241311121

**Published:** 2023-07-05

**Authors:** Sofía Masuelli, Sebastián Real, Patrick McMillen, Madeleine Oudin, Michael Levin, María Roqué

**Affiliations:** 1Institute of Histology and Embryology, National Council of Scientific and Technological Research (CONICET), Parque General San Martin, Mendoza 5500, Argentina; smasuelli@mendoza-conicet.gob.ar (S.M.);; 2Faculty of Medical Science, National University of Cuyo, Parque General San Martin, Mendoza 5500, Argentina; 3Allen Discovery Center, Tufts University, Medford, MA 02155, USA; 4Department of Biomedical Engineering, Tufts University, Medford, MA 02155, USA; 5Faculty of Exact and Natural Sciences, National University of Cuyo, Parque General San Martin, Mendoza 5500, Argentina

**Keywords:** laterality, ion channel, breast cancer, proliferation, stemness

## Abstract

Breast cancer is a complex and heterogeneous disease that displays diverse molecular subtypes and clinical outcomes. Although it is known that the location of tumors can affect their biological behavior, the underlying mechanisms are not fully understood. In our previous study, we found a differential methylation profile and membrane potential between left (L)- and right (R)-sided breast tumors. In this current study, we aimed to identify the ion channels responsible for this phenomenon and determine any associated phenotypic features. To achieve this, experiments were conducted in mammary tumors in mice, human patient samples, and with data from public datasets. The results revealed that L-sided tumors have a more depolarized state than R-sided. We identified a 6-ion channel-gene signature *(CACNA1C*, *CACNA2D2*, *CACNB2*, *KCNJ11*, *SCN3A*, and *SCN3B*) associated with the side: L-tumors exhibit lower expression levels than R-tumors. Additionally, in silico analyses show that the signature correlates inversely with DNA methylation writers and with key biological processes involved in cancer progression, such as proliferation and stemness. The signature also correlates inversely with patient survival rates. In an in vivo mouse model, we confirmed that KI67 and CD44 markers were increased in L-sided tumors and a similar tendency for KI67 was found in patient L-tumors. Overall, this study provides new insights into the potential impact of anatomical location on breast cancer biology and highlights the need for further investigation into possible differential treatment options.

## 1. Introduction

Ion channels (ICH) are transmembrane proteins that regulate the flow of ions across cell membranes, thereby modulating membrane potential, intracellular signaling, and gene expression. Rather than being confined to the nervous systems alone, these abilities are shared by all cells and are considered adaptations of an ancient bioelectric communication system that employed similar strategies to tackle challenges in the past [1]. The ability of cell groups to approach a specific morphology is largely reliant on bioelectric communication across tissues. This information processing occurs through the utilization of identical, well-conserved molecular components like those used by neurons [2]. ICH play crucial roles in many physiological processes, including cell proliferation, migration, invasion, and apoptosis [3]. Dysregulation of ICH can lead to altered membrane potential, intracellular signaling, and gene expression, promoting tumorigenesis and therapy resistance. Any bioelectric disruption can trigger cancerous characteristics [4,5], even in the absence of DNA damage or mutation. Essentially, when cells lose their tight informational connections with neighboring cells, they revert back to a primitive, unicellular state [6], breaking away from the multicellular group and regarding it as a mere external environment. This results in the re-emergence of behaviors appropriate to the unicellular state, such as proliferation. So cancer can be seen as primarily a disorder of establishing informational boundaries between the cellular self and the outside world [7], which has already led to the development of effective treatments that can reverse and prevent cancer in amphibians [8] and is currently undergoing testing in human tissues [9]. Emerging evidence suggests that ICH also participate in the crosstalk between the tumor microenvironment (TME) and cancer cells [10,11], contributing to tumor initiation, progression, and therapy resistance (reviewed in [12,13,14]). For example, ICH expressed by cancer cells can sense and respond to changes in the TME, such as alterations in pH, oxygen tension, and ion concentration, leading to changes in intracellular signaling and gene expression that promote tumorigenesis [3]. Moreover, ICH expressed by TME components, such as immune cells and endothelial cells, can modulate their function and secretion of cytokines and growth factors, affecting tumor cell behavior [14]. Targeting ICH that are involved in TME-tumor interaction may provide new opportunities for developing effective and specific anticancer therapies. 

Breast cancer (BC) is a complex disease that exhibits heterogeneity with diverse subtypes that vary in their molecular and clinical characteristics. One aspect of this heterogeneity is the laterality of breast tumors, i.e., whether the tumor arises in the left (L) or right (R) breast [15]. Although breast tumors are generally thought to be similar regardless of laterality, several studies have suggested that the surrounding tissue of L-R breast tumors may differ [16]. The mammary glands are not identical [17,18,19]; each breast has its unique set of ducts, lobules, and supporting stroma. These structures can differ in size, shape, and branching patterns. In fact, all paired organs present physiological and molecular differences, as shown by microRNA profiling in paired eyes, lung, and testes normal tissues [19]. Therefore, the normal differences between both breasts can lead to potential differences in the microenvironment surrounding tumors impacting tumor biology differently in each breast, thus making it worthwhile to consider when studying BC. 

Interestingly, L-sided BC incidence is slightly increased [20], and even though the underlying causes are unclear [21], the tendency is consistent across populations. Similar differences are observed in other paired organs which laterally differ in cancer incidence and progression [22]. Although these differences seemed not to have a relevant clinical impact [23], growing evidence from research findings indicates they are more important than was considered [24]. For example, in animal studies using mouse models of BC, it was shown that L-R-sided tumors had distinct gene expression profiles [25]. In humans, recent studies discovered that L-R-sided tumors may respond differently to specific therapies depending on the local TME, with tumors in different regions of a single organ showing different expression profiles and properties [26,27]. Furthermore, L-sided BC was found to be associated with a more aggressive biology and a worse outcome as compared to R-BC [28]. Remarkably, recent literature has raised new questions about the relationship between BC laterality and prognosis [29]. Understanding these differences may have significant implications for diagnosis and treatment.

In our previous publications [30,31], we discovered a distinct methylation profile between L-R-sided breast tumors, which we associated with differences in membrane potential and ion concentration. Integrating the facts that L-R normal breast tissues have differences in gene expression and that the bioelectricity is known to play a crucial role in tumorigenesis, we hypothesized that bioelectric differences of both sides of the breast could contribute to the development of BC in a site-specific manner. In our current work, we sought to determine the primary ICH involved in these differences and explore the potential impact on the functional characteristics of the tumor cells. By investigating the role of specific ICH in L-R tumor development, we hope to uncover novel therapeutic targets for BC treatment. 

## 2. Results

### 2.1. Left-Sided Mice Tumors Exhibit Greater Depolarization

We chose to use a mouse model of breast cancer (MMTV-PyMT) which spontaneously develops mammary tumors on both the L and R side, to investigate the role of laterality in regulating the bioelectric properties of tumors. Pairwise comparisons of L-R tumors within the same mouse were possible using this model. After extracting paired tumors, we used two fluorescent probes with opposite charges to measure the bioelectric state establishing a ratio between the fluorescence of both probes (MT/DB). The paired comparison of four tumors, all located on the L-R 4th gland, revealed by confocal microscopy (Figure 1A and Appendix A) a difference in the MT/DB ratio: L-sided tumors showed a decreased MT/DB ratio (Figure 1B and Appendix A). 

In parallel, samples from 3/5 mice were analyzed by flow cytometry, confirming the same trend as seen by microscopy: L-sided tumors had a decreased MT/DB ratio compared to R-sided tumors (Figure 2 and Appendix A). 

This suggests that the L and R glands generate a difference in the bioelectric state of growing tumor cells. The consistent shift into the same direction suggests that L-sided tumors have a more depolarized state than R. 

### 2.2. Bioinformatic Approach: ICH Gene Signature Involved in L-R Bioelectric Differences

Ion channels play a major role in regulating the bioelectrical properties of tumors. To further investigate which ICH may play a relevant role, we switched to a bioinformatic approach. We obtained a list of human ICH genes from the HGNC database. The list contained 330 genes, of which the expression levels of 319 were downloaded from 781 invasive breast tumors in The Cancer Genome Atlas (TCGA) dataset, comprising 393 L-sided and 388 R-sided tumors.

To determine whether a set of ICH genes contributed in a relevant way to the L-R differences, we used the computational method Gene Set Enrichment Analysis (GSEA). We selected the four gene sets that showed significant difference, particularly a consistent decrease of their expression in L-sided tumors (*p* < 0.05; FDR < 0.1): WP-ENERGY-METABOLISM (collection: C2), REACTOME-PHASE-0-RAPID-DEPOLARISATION (collection: C2), REACTOME_CARDIAC-CONDUCTION (collection: C2), and CANCER-MODULES (collection: C4). By combining these gene sets, we identified the leading genes that were associated with the tumor laterality: *CACNA1C*, *CACNA2D2*, *CACNB2*, *KCNJ11*, *SCN3A*, and *SCN3B*, signature that we called: 6-ICH signature.

We first confirmed in the Human Protein Atlas (www.proteinatlas.org, accessed on 16 June 2023) that the six channels had been detected at protein level in normal breast tissue cells (glandular epithelial and mesenchymal cells). Next, we analyzed the expression of the 6-ICH signature in 101 normal vs. 1097 primary breast tumor tissues and in 571 left-sided vs. 526 right-sided breast tumors, using the Xena Functional Genomics Explorer [32] from UCSC (accessed on 16 June 2023). We obtained the expression levels of the signature from an equation where we assumed an equal contribution of each gene, i.e.,
= *CACNA1C* + *CACNA2D2* + *CACNB2* + *KCNJ11* + *SCN3A* + *SCN3B*.

We found a decreased expression of the signature in tumoral cells as compared to normal tissue (Welch’s *t*-test, *p* < 0.001). And among the tumors, we found a decreased expression in L-sided compared to R-sided ones (Mann–Whitney test for non-normal data, *p* = 0.005, Appendix A). Adding more genes to the signature did not significantly increase the association with laterality. However, removing genes one by one resulted in a loss of association. Interestingly, when we compared the expression levels of the individual genes of the 6-ICH signature in a one-by-one manner, we found a reduced or complete loss of association with the side for some genes. Therefore, we considered the 6-ICH signature as robust for further studies.

### 2.3. In Silico: The 6-ICH Signature Inversely Correlates with Pro-Mitotic and Stemness Markers and Survival Rates

ICH play a role in a wide range of biological processes and can regulate opposing cellular phenotypes. Therefore, the 6-ICH signature did not suggest a clearly defined phenotypic impact. To gain insight, we used a bioinformatic approach. We dichotomized the 6-ICH signature expression data into High and Low, using the mean expression value as a cut-off, and performed a differential gene expression (DGE) analysis visualized in a volcano plot. In the Low-expressing group (as compared to the High-expressing one), a significant number of genes were up or down regulated (Figure 3A).

Next, we conducted an enrichment analysis to determine the principal biological processes the DGE were participating in. Interestingly, we found that in the Low-expressing subgroup, mitotic genes were overexpressed, and membrane potential regulating processes were less expressed (Figure 3B). These findings suggest that L-sided tumors with a low-expressing 6-ICH signature may exhibit increased mitotic activity compared to R-sided tumors.

From an in silico cohort of 1082 primary breast tumors from the Xena Functional Genomics Explorer-UCSC, we randomly generated a Discovery Cohort (533 tumors) and a Validation Cohort (549 tumors). We choose the markers “KI67” and “stemness scores” to analyze, as indicative of pro-mitotic behavior, with KI67 being a protein expressed throughout the cell cycle, and stemness scores measuring the extent to which a tissue is capable of self-renewal. So, we studied the correlation between the expression of the 6- ICH signature and the expression of KI67 and an RNA-seq based stemness score [33], both available in the TCGA Pan-cancer Atlas. In the Discovery Cohort, we observed a strong inverse correlation between the 6-ICH signature and both markers, (r = −0.47 and r = −0.62 respectively, Spearman correlation test, *p* < 0.0001) (Figure 4A). Receiver operating characteristic curves (ROC analyses) demonstrated an acceptable discrimination value of the 6-ICH signature to distinguish between high and low expression of KI67 (AUC = 0.74, SE = 0.02, 95% CI = 0.698 to 0.782, *p* < 0.0001) and high or low stemness scores (AUC = 0.79, SE = 0.01, 95% CI = 0.761 to 0.837, *p* < 0.0001). We confirmed these results in the Validation Cohort of 549 primary breast tumors (r = −0.51 and r = −0.62 respectively, Spearman correlation test, *p* < 0.0001; ROC curves for KI67: AUC = 0.74, SE = 0.02, 95% CI = 0.707 to 0.788, and for CD44: AUC = 0.81, SE = 0.01, 95% CI = 0.776 to 0.848, *p* < 0.0001) (Figure 4B). We also confirmed these correlations in a pan-cancer analysis on 9546 tumors (r = −0.51 and r = −0.62 respectively, Spearman correlation test, *p* < 0.0001) (Figure 4C).

We then analyzed the 5-year (1800 days) rates of overall survival, disease-specific survival, disease-free survival, and progression-free survival by comparing high vs. low 6- ICH signature expression in 1094 BC and in 9546 pan-cancer patients. Remarkably, all survival rates were increased in high-expressing tumors (Figure 4D). 

We also examined the correlation between the 6-ICH signature and epigenetic regulators, particularly histone modifiers, methylation erasers, and methylation writers, by performing an expression signature of each group of epigenetic enzymes. We found a strong inverse correlation only between the 6-ICH signature and the signature of DNA methylation writers composed of *DNMT1*, *DNMT3A*, *DNMT3B*, *DNMT3L*, *TRDMT1*, and *DMAP1* (r = −0.42, Spearman correlation test, *p* < 0.0001) (Figure 5 and Appendix A). 

Taken together, these results suggest that the bioelectric signature is associated with the side of the tumor and with cancer hallmarks such as proliferation and stemness markers and is also associated with regulators of DNA methylation (which are known to respond to microenvironmental factors). Moreover, it reveals an impact on the survival rate of cancer patients. The results are robust across different cohorts.

### 2.4. In Vivo: The 6-ICH Signature Inversely Correlates with Proliferation Marker and Stemness Scores

After establishing in silico that the 6-ICH signature was associated with the side and with proliferation and stemness divergences, we progressed to in vivo models in mice to confirm these findings. We utilized paired L-R tumors of four female mice and analyzed the expression of the proliferation marker *KI67* and the stemness marker *CD44*. 

Our analyses revealed a significant increase in the expression of both markers in L-sided tumors compared to R-sided tumors (paired *t*-test, *p* < 0.05) (Figure 6 and Appendix A).

We also investigated whether these differences were present in human breast tumor RNA samples. We analyzed the expression of the markers *KI67* (in 6 L and 5 R) and *CD44* (in 5 L and 4 R) by qPCR. Although not statistically significant, it is notable that we found a similar trend: an increased mean expression of both markers in L-sided tumors (unpaired *t*-test, *p* = 0.2) (Appendix A). 

Taken together, our observations in mice and humans strongly suggest that L-R breast tumors are distinct in two critical cancer hallmarks, namely proliferation and stemness, which are shaped by the differential expression of ICH.

## 3. Discussion

Breast cancer is a heterogeneous disease that encompasses a broad spectrum of molecular subtypes and clinical outcomes [34]. The association between the anatomical location of tumors and their biological behavior is still an open question. Diverse studies across different species have addressed the L-R issue of bilateral organs, like RNA differential profiles in mice paired eyes, lungs, and testes [19], L-R differences acquired during embryonic development [35], L-R metabolic differences in Xenopus [36], in avian [37] and mice [38] models, and in humans [24]. It has been known for many decades that breast tumors occur more frequently in L sides [39], however the causes are still unclear [17,21,24]. In our previous research, we addressed the question of whether epigenetic differences existed between L-R human and mice-xenografted breast tumors. We found differential DNA methylation profiles in genomic regions [30], among which ICH genes were included. Furthermore, we found how L and R extracts from human mammary glands were able to generate membrane potential differences in cultured cells and how L-extract treated cells developed a more depolarized state [31]. So, we proposed that the different L-R environments generated methylation differences which ended up provoking different bioelectric states in the L-R tumors. In this study, we aimed to address the L-R asymmetry question at a whole organism level, in mice and in humans. Our results show that L-sided tumors consistently exhibit a more depolarized state, an increased expression of proliferation and stemness markers, and that the lower level of ICH signature expression in L-tumors is associated with reduced survival rates, as compared to the R-sided ones (Figure 7). 

This suggests that the microenvironments of the L and R sides generate a consistent difference in the bioelectric state of growing tumor cells, with a direct impact on tumor behavior. To our knowledge, prior to our research, there were no published studies informing proliferation differences between L and R breast tumors. However, a recent publication supports our findings, demonstrating (in line with our observations) that L-sided BC is associated with more aggressive biology and worse outcome as compared to R BC. As far as we know, these are the first conclusive findings differentiating breast tumors in bioelectric, proliferative, and survival rate terms, based on their side.

We identified a 6-ICH gene expression signature that was consistently associated with the tumor side. The signature is composed of 3 calcium, 2 sodium, and 1 potassium channels. CACNA1C, CACNA2D2, and CACNB2, are all subunits of voltage-gated calcium channels, which are involved in a wide range of physiological processes, including muscle contraction, neurotransmitter release, and gene expression [40]. *KCNJ11*, on the other hand, codes for a potassium channel that is involved in regulating insulin secretion in pancreatic beta cells [41]. Bioelectric signaling driven by potassium channels has been proposed as metastasis regulators in triple-negative breast cancer [42]. Finally, SCN3A and SCN3B are subunits of voltage-gated sodium channels, which are important for the initiation and propagation of action potentials in neurons and other excitable cells [43]. Recent findings have revealed the clinical importance of sodium channels and the ionic microenvironment of breast tumors [44]. Cell cycle checkpoints, which rely on being heavily dependent on bioelectricity, have been linked to errors that occur when the number or current of sodium channels increases [45].

Predicting the outcome of a gene expression shift in these six ICH entities, however, is challenging. For example, reduced expression of *CACNA1C*, *CACNA2D2*, and *CACNB2* in cardiac myocytes could result in decreased calcium influx and reduced contractility, which could lead to a hyperpolarized state. Additionally, reduced expression of *SCN3A* and *SCN3B* in neurons could result in decreased sodium influx and reduced excitability, which could also lead to a hyperpolarized state. However, reduced expression of *KCNJ11* in pancreatic beta cells could lead to decreased potassium efflux and increased insulin secretion, which could result in a depolarized state. Overall, the impact of changed expression of ICH on membrane potential and cellular excitability is complex and context-dependent and requires consideration of the specific ion channel and cell type involved. Therefore, the 6-ICH signature did not suggest a clearly defined phenotypic impact. However, what was clear was the observation that the distinctive L/R expression had an influence on membrane potentials. This phenomenon can be attributed to the fact that the membrane potential is reliant on the collective actions of ion channels and transporters operating at the cellular membrane. Our findings indicate that the less the signature expression, the more depolarized is the membrane potential. Speculating on the function of the genetic signature channels, one could propose in general terms that less function of voltage independent potassium channels (like KCNJ11) would result in membrane depolarization, which would trigger the activation of voltage gated calcium and sodium channels. This sequential process would maintain a more depolarized membrane potential involved in cell cycle progression. 

Cancer cells are known to tend to be more depolarized than their normal counterparts [46]. Evidence from both rodent and human tissues has shown that fast proliferating tumor cells exhibit a depolarized membrane potential, whereas non- or less-proliferating are distinguished by their hyperpolarized membrane potential [47]. This has allowed it to be stated that depolarization serves as a signal to initiate mitosis and DNA synthesis, independently of the fluctuation of membrane potential across the cell cycle [47]. Membrane potential has also a role in normal stem cell differentiation; in particular, a hyperpolarization is required during cell maturation, while on the contrary, depolarization reduces the differentiated phenotype [48].

By bioinformatic analyses in TCGA breast tumors, the lower 6-ICH expression was revealed to be associated with increased proliferation and stemness, and with the worst survival in 5 years. Proliferation and stemness are key biological processes in tumorigenesis. Importantly, higher values for stemness indices are associated with an increased number of cancer stem cells and with a greater de-differentiated phenotype [33]. Stemness is defined as the potential of a cell for self-renewal and differentiation. Cancer progression involves gradual loss of differentiated phenotype and acquisition of stem cell-like features. Less differentiated tumors are more likely to spread to distant organs and make prognosis worse, particularly because metastases are usually resistant to available therapies [49]. Proliferation is also a critical aspect to consider when evaluating BC, and *KI67* expression stands out as one of the most important and cost-effective surrogate markers for assessing tumor cell proliferation [50]. Our results suggest that the 6-ICH signature may play a role in modulating these processes in breast cancer, probably by changing the bioelectric state in a side-dependent manner. 

The TME is a complex and dynamic ecosystem that plays a critical role in tumor initiation, progression, and response to therapy. It comprises various cell types, extracellular matrix components, and signaling molecules that interact with tumor cells in a bidirectional manner [51]. Growing evidence suggests that the TME can alter the epigenetic landscape of cancer cells, leading to changes in gene expression patterns that promote tumorigenesis and therapy resistance. For example, TME-derived factors, such as growth factors, cytokines, and extracellular matrix proteins, can activate oncogenic signaling pathways that promote DNA methylation, histone modifications, and chromatin remodeling in cancer cells [52]. Moreover, TME components, such as cancer-associated fibroblasts, immune cells, and endothelial cells, can secrete exosomes and micro-vesicles that transfer epigenetic modifiers, such as micro-RNAs, DNA methyltransferases, and histone deacetylases, to cancer cells, further altering their epigenetic landscape [10,11]. The impact of TME on the tumoral epigenome and its effects on tumor behavior are of great interest because epigenetic changes are reversible and potentially targetable by therapeutic agents. 

Even though not addressed in this study, it is important to consider here that lipids play a crucial role in the cellular membrane potential during cancer processes. Alterations in lipid metabolism and membrane composition are commonly observed in cancer cells, with significant implications for the membrane potential. Lipids, such as phospholipids, influence membrane fluidity and impact the activity of ion channels and signaling pathways [53]. Lipid rafts, specialized regions enriched with cholesterol and sphingolipids, organize membrane proteins involved in signaling and affect ion channel localization and activity [54]. Certain lipid molecules participate in signaling cascades that regulate the membrane potential [55]. Altered lipid metabolism in cancer cells can affect the composition and properties of the cell membrane, influencing ion channel function and membrane potential. Understanding the interplay between lipids and membrane potential offers valuable insights for research and potential therapeutic interventions.

Our finding that differences between L-R breast tumors are related to ICHs suggests possible therapeutic opportunities by applying already approved drugs used in other diseases to modulate channels. Dysregulation of ICH has been implicated in various diseases. In recent years, ICH modulating drugs have gained attention as potential cancer therapeutics. Drug repositioning is a strategy that involves the repurposing of existing drugs for new therapeutic indications. In the context of cancer treatment, repositioning ICH modulating drugs has emerged as a promising approach to identify new treatments for cancer [56] and the use of ICH drugs in cancer is increasingly being studied, as reviewed in [57]. An oncology-focused drug repurposing database has been developed, comprising drugs that show potential for repurposing in cancer treatment (https://www.anticancerfund.org/en/redo-db, accessed on 4 May 2023). Among the 369 drugs listed to date, there are 20 that target some of the six channels included in our 6-ICH signature, which sounds very encouraging. By repositioning these drugs for cancer treatment, the development timeline and costs associated with drug development can be reduced, making it an attractive option for drug discovery. 

Finally, another evidence of our study is that the ICH signature was found inversely associated with DNMT expression, and not with other epigenetic modulators (e.g., histone deacetylases, acetylases, histone methyl transferases, DNA methyl-erasers). It is now well established that epigenetic dysregulations play pivotal roles in cancer onset and progression, including DNA methylation. To occur, these epigenetic events need to be triggered by genetic alterations or transcription shifts of the epigenetic regulators. In this, the TME plays a fundamental role. Therefore, it is valid to reason that the ICH signature is regulated by DNA methylation, and further studies should be performed to confirm this, by for example inhibiting in vitro DNMTs function with 5-azacytidine or deoxycytidine.

In conclusion, our study suggests that the anatomical location of breast tumors may influence their bioelectric state, potentially mediated by ICH. The 6-ICH signature identified in our study may play a role in modulating key biological processes involved in cancer progression, and further studies are warranted to validate these findings. The potential impact of anatomical location on BC biology and clinical outcomes should also be further investigated, suggesting differential treatment options with repurposed ICH drugs.

## 4. Methods

### 4.1. Human Primary Breast Tumors

RNA and tumor tissue from our patient’s tumor tissue bank was used for this study. Informed consent had been originally obtained from all subjects’ tissue included in the bank [58]. The study was conducted according to the guidelines of the Declaration of Helsinki and approved by the Ethics Committee of the School of Medical Sciences, National University of Cuyo, Mendoza, Argentina. (Protocol code: 15856/2016, date of approval: 16 September 2016).

### 4.2. Mouse Model MMTV-PYMT

All animal procedures were reviewed and approved by the Tufts University Institutional Animal Care and Use Committee. Five MMTV-PyMT (002374, Jackson Laboratory, Bar Harbor, ME, USA) female mice were housed together in the same cage in an on-site housing facility, with access to standard food, water, and a 12/12 light cycle. At 13 weeks of age, the MMTV-PyMT mice were euthanized using CO_2_, and paired L-R breast tumors from 5 mice, all from the 4th gland, were excised. Tumors were fractionated into 5 pieces for measuring membrane potential by confocal microscopy and flow cytometry, and for DNA and RNA extraction. 

### 4.3. Membrane Potential Measured by Flow Cytometry

Tumors were desegregated in a Petri dish with 1–2 mL of disaggregation solution containing 1 mL collagenase 2 mg/mL in PBS, 150 µL of hyaluronidase 10 mg/mL in PBS (Sigma, #H4272, St. Louis, MO, USA), 100 µL antibiotic (penicillin-streptomycin-glutamine 100×, Gibco, #10378016, Life Technologies, Carlsbad, CA, USA) and 8.75 mL of DMEM/F12 media (Gibco, #21041025, Life Technologies, Carlsbad, CA, USA) with FBS 2% (SH30071.03, Cytiva, Marlborough, MA, USA). Tumors were cut into small pieces using vesture and all the remaining fat/normal mammary gland/skin was removed. The mix of tumor tissue and disaggregation solution was transferred to a tube and digested at 37 °C with agitation for 2–4 h. Cells were filtered with 40 µm cell strainers, then centrifuged for 5 min at 10,000 rpm and washed 4 times with cold PBS. Afterwards, cells were resuspended in 200 µL DMEM FluoroBrite (Gibco, #A1896701, Life Technologies, Carlsbad, CA, USA) with 2% FBS. DiBAC4(3) 2 µM (positively charged, DB) (Invitrogen, #B438, Life Technologies, Carlsbad, CA, USA) and Mytotracker Deep Red FM 500 nM (negatively charged, MT) (Cell Signaling Technology, #8778, Danvers, MA, USA) was added and incubated for 30 min at 37 °C. We used the MT probe instead of rhodamine 6G (which is a potentiometric dye [59]), because they have similar functionality and to preclude rhodamine 6G’s spectral overlap with DIBAC4(3). Fluorescence was measured by flow cytometry (Attune NxT flow cytometer^®^) using a 530/30 emission filter for DB measurement and a 670/14 emission filter for MT measurement. The autofluorescence of each tumor cell was measured and subtracted. Results were analyzed using FlowJo v X.0.7^®^ software 10.8.1 (RRID: SCR_008520). 

### 4.4. Membrane Potential Measured by Confocal Microscopy 

Thin slices of the paired L-R 4th gland tumors were cut and placed in Petri dishes with DMEM FluoroBrite (Gibco, #A1896701, Life Technologies, Carlsbad, CA, USA) plus 2% FBS. For measuring membrane potential, 2–3 slices per tumor were used, incubating 15 min at 37 °C with media containing DB 2 μM and MT 500 nm in Fluorobrite DMEM. One slice was used to measure autofluorescence of the cells. Images were captured using a Leica Stellaris Sp8 confocal microscope at 10× magnification, at 37 °C and 3–5 images were taken per tumor, with 3 z-slices per image. As control for normalized measurements, cells were treated with a high concentrated KCl (65 mM) solution as depolarizing agent, for 15 min at 37 °C. Afterwards, slices were incubated with Fluorobrite DMEM containing DB (2 μM) and MT (500 nM). The images were processed using Image J software (ImageJ 1.53t/Java 1.8.0_322, National Institutes of Health, USA, RRID: SCR_003070). The sum of the z-slices was used for image projection. A ratio of MT/DB integrated density was calculated for each image, and this ratio was normalized to the same ratio obtained from the depolarized control images. 

### 4.5. RNA Extraction

RNA was extracted from mice tumors using the Quick-RNA Miniprep Plus Kit (Zymo Research, #R1057, Irvine, CA, USA), following the manufacturer’s protocol. RNA from human breast tumors was extracted with a Trizol (Life Technologies, Carlsbad, CA, USA) based protocol. 

### 4.6. Gene Expression Analysis in Mice Tumors by Droplet Digital PCR (ddPCR)

Purified RNA from mice tumors was converted to cDNA using M-MLV Reverse Transcriptase (Inbio #K1600, Tandil, Buenos Aires, Argentina), and cDNA was stored at −20 °C until use. For each reaction, 1 ng was used plus ddPCR Supermix for Probes (No dUTP) (BioRad, #1863023, Hercules, CA, USA) and PrimeTime^®^ Mini qPCR Assay probes for *KI67*, *CD44* and *HPRT* (housekeeping) genes with FAM fluorescence for detection. Sequences for *KI67* probe: 5′-/56-FAM/TGGCCTACC/ZEN/TGGTCTTAGTTCCGT/31ABkFQ/-3′, primer 1: 5′-TTCCTTCAGCAAGCCTGAG-3′, primer 2: 5′-CTTCATAGGCATTCCCTCACTC-3′. For *CD44* probe: 5′-/56-FAM/ACCCATACC/ZEN/TGCATGTTTCAAAACCC/31ABkFQ/-3′, primer 1: 5′-GCTTTCAACAGTACCTTACCCA-3′, primer 2: 5′-GGATGAATCCTCGGAATTACCA-3′. For *HPRT* probe: 5′-/56-FAM/CTTGCTGGT/ZEN/GAAAAGGACCTCTCGAA/31ABkFQ/-3, primer 1: 5′-CCCCAAAATGGTTAAGGTTGC-3′, primer 2: 5′-AACAAAGTCTGGCCTGTATCC-3′. We used the following probe concentrations to improve the multiplexed reaction: probe *CD44* 1.5X with *HPRT* 1X; and probe *KI67* 1X with *HPRT* 2X. For each assay, droplets were generated using droplet generation oil for probes (Bio-Rad, #1863005, Hercules, CA, USA) on the QX200 Droplet Generator (Bio-Rad, #17005227, Hercules, CA, USA) according to the manufacturer’s protocol adding the specific primers. Droplets were cycled on the C1000 Touch Thermal Cycler (Bio-Rad, Hercules, CA, USA) for 40 cycles, with a 58 °C annealing temperature. Droplets were read using the QX200 Droplet Reader (BioRad, Hercules, CA, USA). Data were analyzed with QuantaSoft software version 1.7 (BioRad, Hercules, CA, USA). 

### 4.7. Gene Expression Analysis in Human Tumors by qPCR

One µg of total RNA was used for the synthesis of cDNA using M-MLV retro-transcriptase (Inbio #K1600, Tandil, Buenos Aires, Argentina) and Random Hexamers (Qiagen #79236, Germantown, MD, USA) primers. An amount of 15 ng was used to perform Real-Time PCR using specific primers for *CD44*, *KI67* and *β-actin* (housekeeping) in an AriaMx Real-time PCR System (Agilent, Santa Clara, CA, USA). The sequences of the primers for *CD44* were as follows: forward 5-TGCCGCTTTGCAGGTGTATT-3 and reverse 5-CCGATGCTCAGAGCTTTCTCC-3; for *KI67*: forward 5-TGACCCTGATGAGAAAGCTCAA-3 and reverse 5-CCCTGAGCAACACTGTCTTTT-3; for *β-actin*: forward 5-TGACGTGGACATCCGCAAAG-3 and reverse 5-CTGGAAGGTGGACAGCGAGG-3. For the reaction, a Master Mix qPCR 2X with SYBR was used (Inbio, Tandil, Buenos Aires, Argentina). The amplification program consisted of 40 cycles of 15 s at 95 °C, 30 s at 60 °C, and 30 s at 72 °C, followed by a final melting curve step. The analysis was performed using AriaMx Software version 2.0 (Agilent, Santa Clara, CA, USA). Relative expression normalization of genes of interest was carried out using β-actin gene expression as endogenous reference control by the ΔCq method.

### 4.8. Public Datasets and Platforms Used

Human ICH genes were obtained from the HUGO Gene Nomenclature Committee (HGNC) database (https://www.genenames.org/data/genegroup/#!/group/177, accessed on 10 February 2023). 

Gene expression and laterality data of primary breast tumors were downloaded from The Cancer Genome Atlas (TCGA)-Xena Functional Genomics Explorer-UCSC (https://xenabrowser.net/, accessed on 16 June 2023) (TCGA-Breast Cancer dataset) and exploration tools for differential gene expression (DGE), enrichment analyses, and Kaplan-Meier curves were used. 

Dataset enrichment score analyses to compare L-R tumors were performed using the computational method gene set enrichment analysis (GSEA). GSEA allows it to be determined whether pre-defined gene sets show significant differences between two phenotypes (in this case, L- and R-sided tumors). Based on gene expression data from these two biological states, GSEA calculates and ranks the DGE, searches for the gene sets in the Human Molecular Signature Database (MSigDB) containing 33,591 gene sets and identifies the top ones that significantly contribute to the differences between the phenotypes. Leading-edge analysis using the GSEA tool, was used to find associated gene signatures.

### 4.9. Statistics

Statistical comparisons were performed using GraphPad Prism version 5.03 for Windows, GraphPad Software (San Diego, CA, USA, www.graphpad.com). Un-paired or paired Student’s *t*-test were used according to the experiment type, to compare the means of the fluorescence ratios. Data from different experiments were normalized to a maximum depolarized state by using high concentrated KCl solution. All results are means of 3 independent experiments with 2–3 technical replications each. Correlation analyses between two variables were performed using Spearman rank coefficient calculations. Sensitivity/specificity values of the 6-ICH signature to distinguish between high vs. low proliferation and stemness markers were calculated using the receiver operating characteristic curve (ROC). Overall survival, disease free survival, progression free survival, and disease specific survival rates were analyzed by Kaplan-Meier plots and *p* values < 0.05 were considered statistically significant, using FDR correction (<0.1) when needed for repetitive comparisons. 

## Figures and Tables

**Figure 1 ijms-24-11121-f001:**
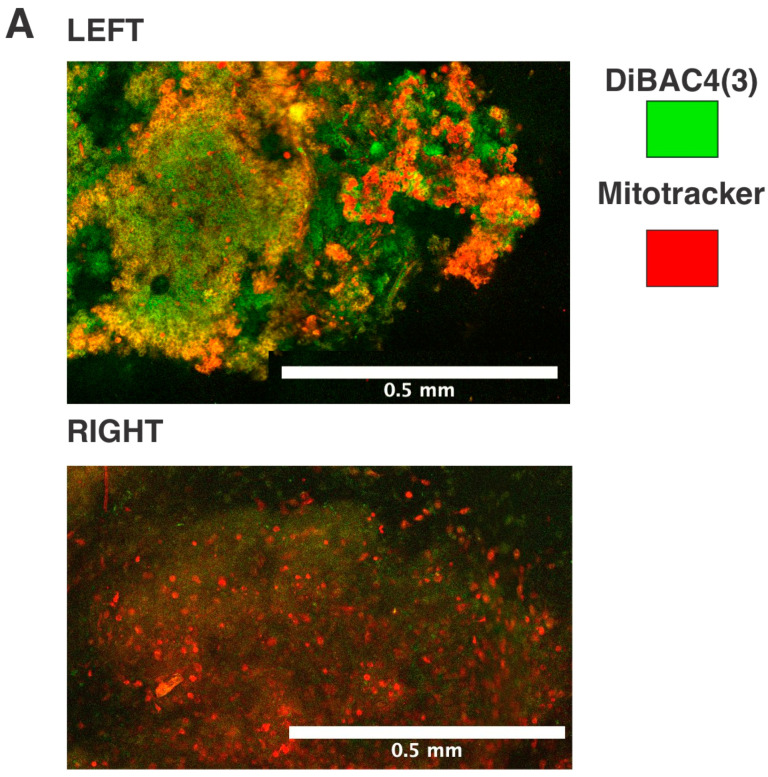
Differential MT/DB fluorescence ratio in L-R mice breast tumors. (**A**) representative confocal microscopy image of L (upper panel) and R (lower panel) of paired mouse breast tumor tissue, stained with MT (red) and DB (green) fluorescent probes. Fluorescence gradients of each fluorophore are shown. (**B**) Quantification of MT/DB ratio in 4 pairs of L-R mice breast tumors, as measured in different images by fluorescence microscopy. The L-mean of MT/DB fluorescence is lower than the R-mean (unpaired *t*-test, *p* = 0.01). Circles = Mouse 2, Triangles = Mouse 3, Squares = Mouse 4, Inverted Triangles = Mouse 5, * = *p* value between 0.01 and 0.049.

**Figure 2 ijms-24-11121-f002:**
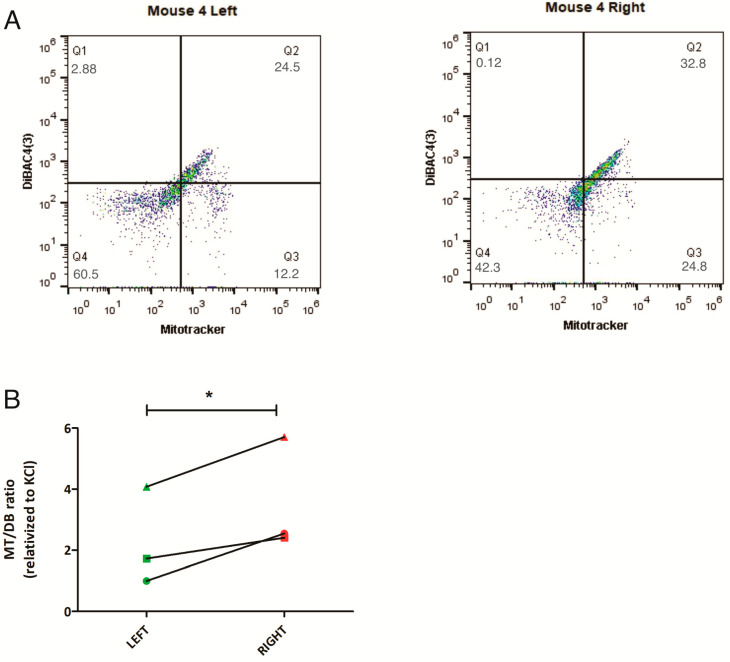
Differential MT/DB fluorescence in L-R mice breast tumors. (**A**) representative flow cytometry image of breast tumor tissue, stained with MT and DB fluorescent probes. Left panel = L tumor cells, Right panel = R tumor cells. As can be seen, L-tumor cells show a decreased Q3 (% of MT-stained cells)/Q2 (% of DB-stained cells) ratio, as compared to R-cells. Green dots represent increased number of cells, whereas blue dots show a smaller number of cells. (**B**) quantification of MT/DB ratio in R-L tumor cells in 3 mice: mouse 1 (represented as circle), mouse 4 (as square), and mouse 5 (as triangle). In all, L-tumor cells show decreased MT/DB ratio (paired *t*-test, *p* = 0.05). * = *p* value between 0.01 and 0.049.

**Figure 3 ijms-24-11121-f003:**
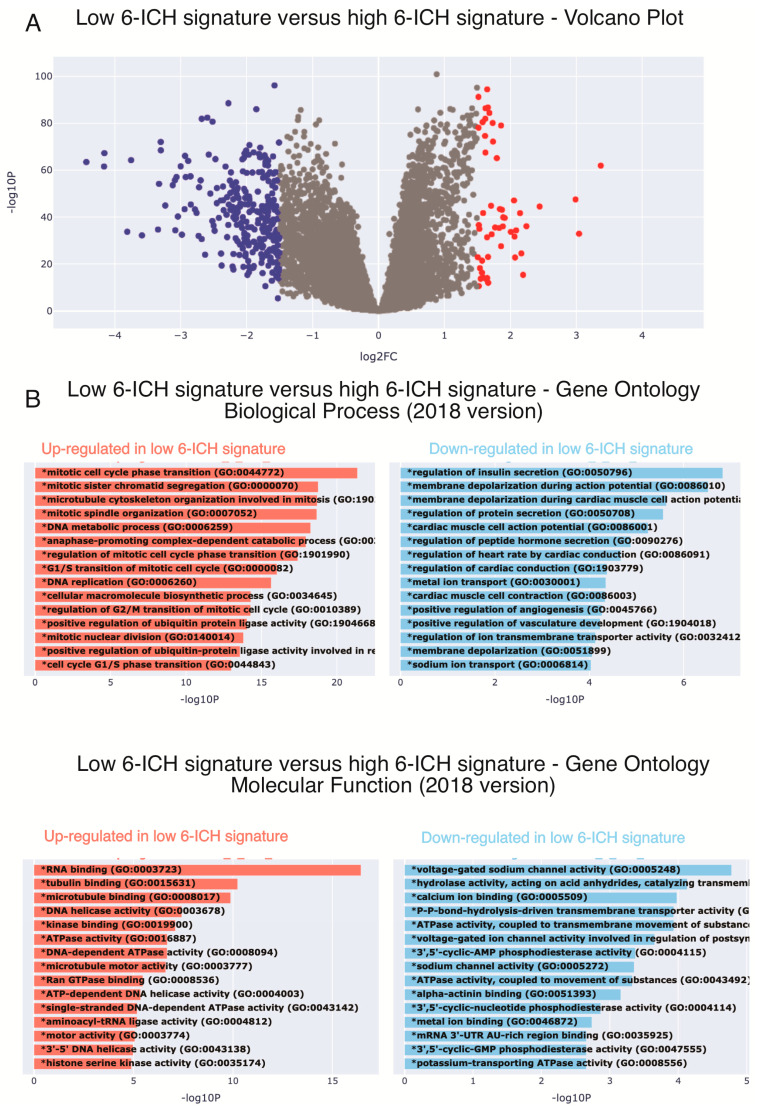
Low expression of the 6-ICH signature associates with pro-mitotic pathways. (**A**) Differential gene expression between Low vs. High expression of the 6-ICH signature. Volcano plot showing the significantly down-regulated (blue dots) and up-regulated (red dots) genes in the Low-expressing subgroup, as compared to the High-expressing one. Performed in Xena Functional Genomics Explorer-USCS (https://xenabrowser.net/, accessed on 1 June 2023) (*p* values < 0.05). (**B**) Enrichment analyses of the DGE between Low-expressing vs. High-expressing samples. Pro-mitotic biological processes and molecular functions (2018 version) are over-expressed in the Low-expressing subgroup. * = *p* value between 0.01 and 0.049.

**Figure 4 ijms-24-11121-f004:**
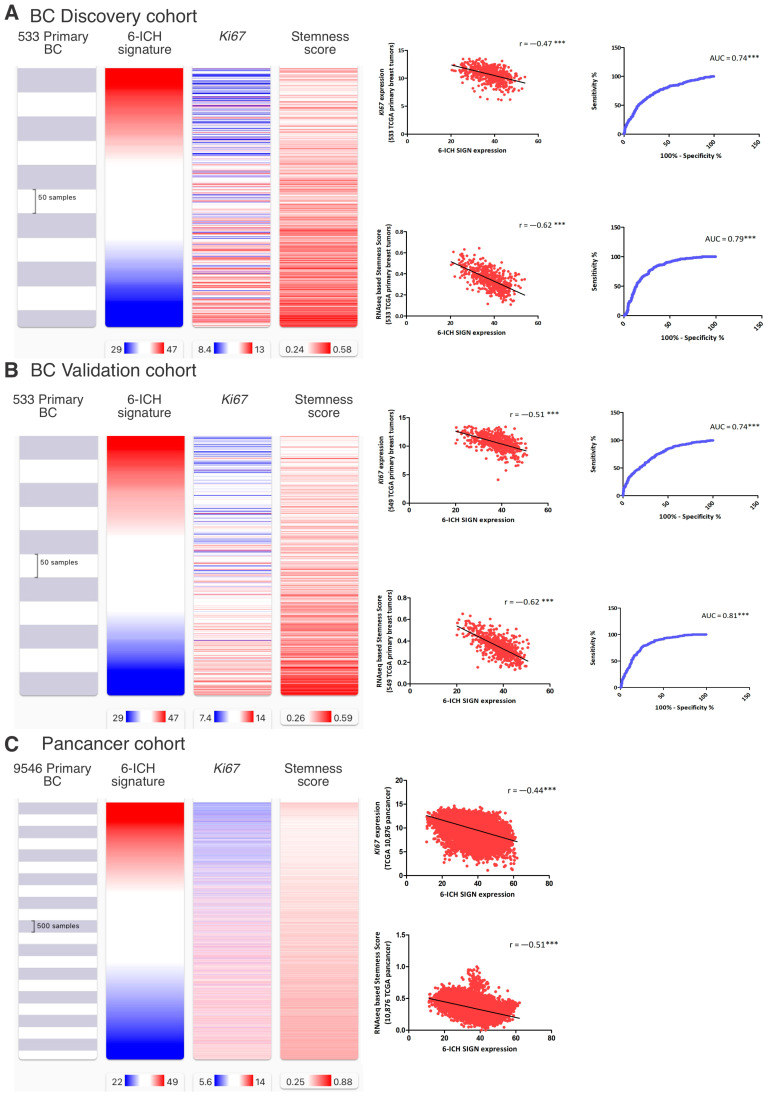
Inverse correlation of the 6-ICH signature with proliferation, stemness markers, and survival rates. (**A**,**B**) Expression data in a Discovery Cohort of 533 primary breast tumors and a Validation Cohort of 549 primary breast tumors. Blue indicates less expression, red increased expression. Inverse correlation is shown between 6-ICH signature expression and markers KI67 and stemness score; ROC curves are presented showing the sensitivity and specificity of the 6-ICH signature to distinguish high from low KI67 and stemness score. (**C**) Expression data of the 6-ICH signature in a Pan-cancer cohort of 9546 tumors of different types. (**D**) Kaplan-Meier curves showing decreased survival (blue curves) in patients with tumors expressing lower 6-ICH signature, in a breast cancer and a pan-cancer cohort. *** = *p* value < 0.0001.

**Figure 5 ijms-24-11121-f005:**
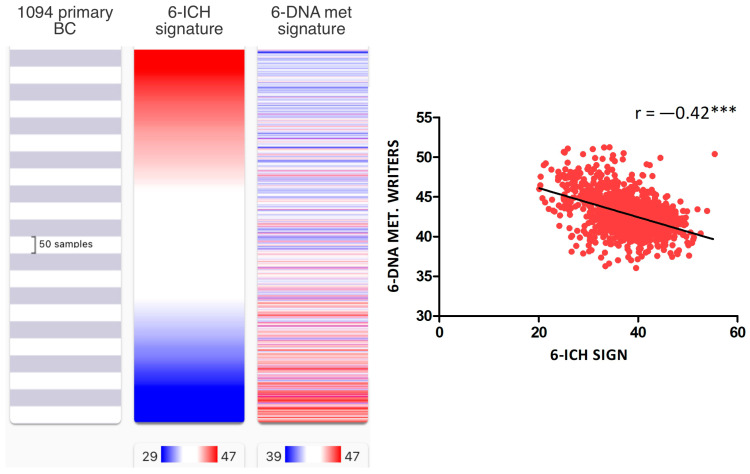
Inverse correlation of the 6-ICH signature with DNA methylation writers’ expression. In 1094 primary BC from TCGA, inverse correlation between the 6-ICH signature and the signature of DNA methylation writers composed of *DNMT1*, *DNMT3A*, *DNMT3B*, *DNMT3L*, *TRDMT1*, and *DMAP1* (r = −0.42, Spearman correlation test, *p* < 0.0001). Blue indicates less expression, red increased expression. *** = *p* value < 0.0001.

**Figure 6 ijms-24-11121-f006:**
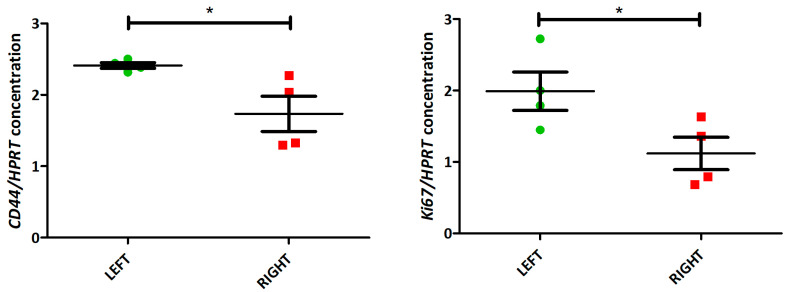
Increased expression of proliferation and stemness markers in L-sided mice tumors. *KI67* and *CD44* expression in 4 paired L-R mice tumors (by ddPCR). Both markers are increased in L-tumors (paired *t*-test, *p* < 0.05). * = *p* value between 0.01 and 0.049.

**Figure 7 ijms-24-11121-f007:**
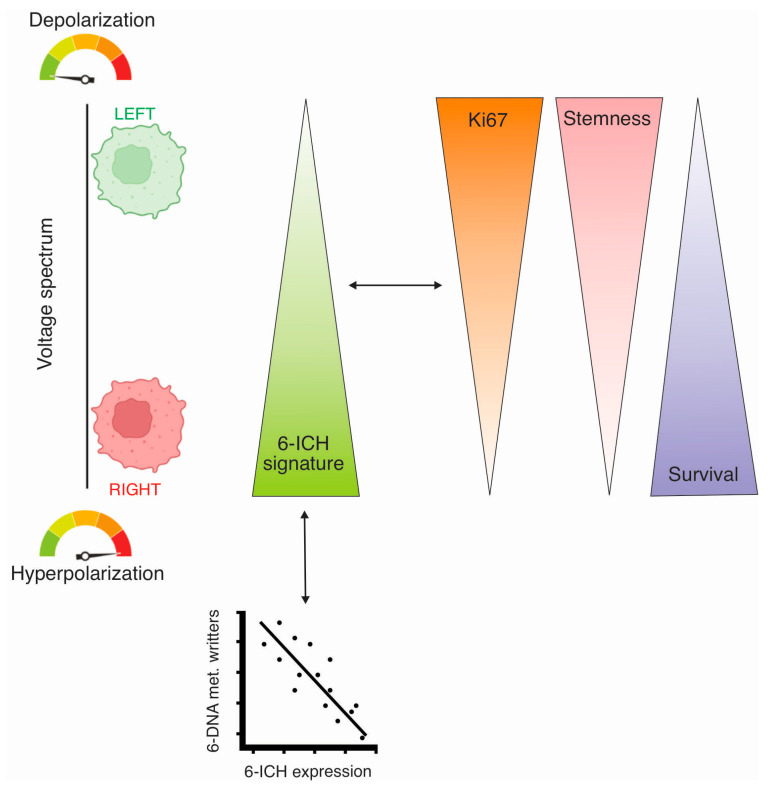
Schematic abstract of the results of the study, performed by Biorender. Left breast tumors were found more depolarized, with less expression of a 6-ICH gene signature, associated with increased *KI67* and stemness markers expression, and presenting decreased survival time, as compared to right-sided tumors. Additionally, the ICH signature showed an inverse correlation with the expression of a 6-DNA-methylation writers’ signature. Members of the 6-ICH signature: *CACNA1C*, *CACNA2D2*, *CACNB2*, *KCNJ11*, *SCN3A*, *SCN3B*. Members of the DNA methylation writers: *DNMT1*, *DNMT3A*, *DNMT3B*, *DNMT3L*, *TRDMT1*, *DMAP1*.

## Data Availability

No new data were created in this study. Data sharing is not applicable to this article.

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
