# Peer review of "The Yin and Yang of Breast Cancer: Ion Channels as Determinants of Left–Right Functional Differences"

_ijms, 2023, doi:10.3390/ijms241311121_

Round 1
Reviewer 1 Report
The authors demonstrated that the anatomical location of breast tumors may influence their bioelectric state, potentially mediated by ICH. They identified e 6-ICH signature that may play a role in modulating key biological processes involved in cancer progression.
They can add a schematic figure of the work with the main results.
In line 54 they must add the full name of the TME acronym.
In line 319 they must remove the full name of the TME acronym.
In figure 1A, they must add the magnification and/or the scale bar.
In figure 2B, can they indicate the significance for each mice?
In the discussion, they can improve the part of CACNA1C, CACNA2D2, CACNB2, KCNJ11, SCN3A and SCN3B and role in tumors.
Author Response
To Reviewer 1
- They can add a schematic figure of the work with the main results.
We thank the reviewer for this observation. A schema of the main results has been included as Figure 7 in the Discussion section (line 347), with its respective legend. We consider that he integration of the results is visualized better now and allows for a clear understanding of the scope of the work.
- In line 54 they must add the full name of the TME acronym.
Done
- In line 319 they must remove the full name of the TME acronym.
Done
- In figure 1A, they must add the magnification and/or the scale bar.
Done.
- In figure 2B, can they indicate the significance for each mice?
We have changed the graphic style of Figure 2B to a scatter plot, where the symbol shapes refer to the 3 different mice (line 159). We consider this change has improved the comprehension of the result. Paired T-test has been performed comparing means, instead of two-way anova as in the previous version of the manuscript, showing the same p-value and result.
- In the discussion, they can improve the part of CACNA1C, CACNA2D2, CACNB2, KCNJ11, SCN3A and SCN3B and role in tumors.
The Discussion section has been improved. We have included two explicative paragraphs, in which we discuss the potential impact of the channel signature on tumoral hallmarks. (lines 385-401). In addition we have expanded information about the role in cancer of the individual channels and included the respective references (49, 50 and 51).

Reviewer 2 Report
The manuscript "The Yin and Yang of Breast Cancer: Ion Channels as Determinants of Left-Right Functional Differences" is nicely written, original, and brings results obtained with different methods.
I have a few suggestions to make:
1) Ion channels may be expressed at the mRNA level but not at the protein level. Moreover, the traffic to the membrane may be impeded in tumoral cells, as proved in many previous papers. I recommend western blot analysis to confirm these proteins' presence in the breast cancer cell's membrane.
2) Ion channel may be present but not functional. Patch-clamp or voltage-clamp experiments would improve the results.
3) Membrane potential is modulated by ion channels and lipids. In breast cancer cells is documented that membrane composition is altered and some lipids were proposed as cancer biomarkers. If it is too difficult to quantify the lipids from R/L tissues, I recommend at least a short discussion about this possibility.
Author Response
To Reviewer 2:
- Ion channels may be expressed at the mRNA level but not at the protein level. Moreover, the traffic to the membrane may be impeded in tumoral cells, as proved in many previous papers. I recommend western blot analysis to confirm these proteins' presence in the breast cancer cell's membrane.
We do completely agree with the reviewer that mRNA is not definitive. Upon the reviewer's request, we aimed to identify the expression of the 6-ICH signature at protein level. Due to our limited access to normal human breast tissue and given in addition that it is beyond our economic reach to perform Western blots of the 6 channels, we conducted a search using tissue data from The Human Protein Atlas (https://www.proteinatlas.org/). By this, we could confirm that the 6 proteins are present in the normal mammary glandular epithelium (information which we have included now in the Result section line 185). Also, a relevant information previously omitted in the earlier manuscript version, is that the expression of the genetic signature DECREASES in tumor tissue when compared to normal tissue (which is reasonable since the decrease is associated with a depolarized state, a characteristic trait of tumor cells). So also, this has now been included in the Results section of this new manuscript version (lines 192).
When evaluating the possibilities to perform a few WB on tumor tissue, considering the decrease of expression, we suspect that the detection might be unreliable not only due to potential antibody-related failures but also due to the tumor biology itself. Therefore, we thought that negative WB results would not greatly prove nor contribute much to the hypothesis. Furthermore, considering that our study includes primarily bioelectrical observations, we think that these physiological results are showing the final endpoint.
- Ion channel may be present but not functional. Patch-clamp or voltage-clamp experiments would improve the results.
We do agree with the reviewer that voltage-clamp experiments would be outstanding. However, it's not known if these cells be readily patched. Therefore, DiBAC has been used. In prior studies of two co-authors of this manuscript, DIBAC has been calibrated to patch clamp in breast cancer cell lines (Bonzanni, M.; Payne, S.L.; Adelfio, M.; Kaplan, D.L.; Levin, M.; Oudin, M.J. Defined Extracellular Ionic Solutions to Study and Manipulate the Cellular Resting Membrane Potential. Biol. Open 2020, doi:10.1242/bio.048553) so we propose it as a reliable indicator (especially as we do it in this study:i.e. not measuring absolute voltage, but relative Left-Right difference). For studies primarily conducted in countries like Argentina, where routine access to patch-clamp approaches is limited, the utilization of calibrated surrogates becomes highly valuable in advancing scientific findings.
- Membrane potential is modulated by ion channels and lipids. In breast cancer cells is documented that membrane composition is altered and some lipids were proposed as cancer biomarkers. If it is too difficult to quantify the lipids from R/L tissues, I recommend at least a short discussion about this possibility.
In the Discussion section, a paragraph has been included discussing the role of lipids in the membrane potential, through alterations in the membrane structure, molecular signaling and lipids’ metabolism (line 432-443) including the respective references (57, 58 and 59).

Round 2
Reviewer 2 Report
I agree with the manuscript in this form.
The manuscript is easily readable; the English is good for publication.